# The Effect of Substrate Temperature on the Evaporative Behaviour and Desiccation Patterns of Foetal Bovine Serum Drops

**Marina Efstratiou** [1] , **John Christy** [1,*] , **Daniel Bonn** [2] and **Khellil Sefiane** [1]

1   Institute of Multiscale Thermofluids, School of Engineering, The University of Edinburgh, James Clerk Maxwell Building, Peter Guthrie Tait Road, King's Buildings, Edinburgh EH9 3FD, UK; Marina.Efstratiou@ed.ac.uk (M.E.); K.Sefiane@ed.ac.uk (K.S.)
2   Institute of Physics, University of Amsterdam, Science Park 904, 1098 XH Amsterdam, The Netherlands; D.Bonn@uva.nl
*   Correspondence: J.Christy@ed.ac.uk

**Abstract:** The drying of bio-fluid drops results in the formation of complex patterns, which are morphologically and topographically affected by environmental conditions including temperature. We examine the effect of substrate temperatures between 20 °C and 40 °C, on the evaporative dynamics and dried deposits of foetal bovine serum (FBS) drops. The deposits consist of four zones: a peripheral protein ring, a zone of protein structures, a protein gel, and a central crystalline zone. We investigate the link between the evaporative behaviour, final deposit volume, and cracking. Drops dried at higher substrate temperatures in the range of 20 °C to 35 °C produce deposits of lower final volume. We attribute this to a lower water content and a more brittle gel in the deposits formed at higher temperatures. However, the average deposit volume is higher for drops dried at 40 °C compared to drops dried at 35 °C, indicating protein denaturation. Focusing on the protein ring, we show that the ring volume decreases with increasing temperature from 20 °C to 35 °C, whereas the number of cracks increases due to faster water evaporation. Interestingly, for deposits of drops dried at 40 °C, the ring volume increases, but the number of cracks also increases, suggesting an interplay between water evaporation and increasing strain in the deposits due to protein denaturation.

**Keywords:** foetal bovine serum; drops; evaporation; drying; proteins; bio-colloids; pattern formation; cracking; substrate temperature

## 1. Introduction

The study of the 'coffee ring' in evaporation of drops containing suspensions or soluble solids in the late 1990s [1], triggered wide interest in studying sessile drop evaporation of both pure [2,3] and complex fluids [4–10], as well as investigation of the patterns arising from drops desiccation [11–13]. For colloidal suspensions, particles are carried to the contact line by an outward flow developing due to continuity, leading to the formation of a ring-like deposit. During drying, various hydrodynamic and physicochemical processes occur [14], including droplet spreading and adhesion to the substrate, contact line pinning, internal flows, crystallisation and cracking [15,16]. As a result, complex desiccation patterns form, with various morphological characteristics such as rings, crystalline structures and cracks [11,17–20].

Biological fluids are considered to be complex colloidal systems, consisting mainly of proteins, electrolytes and water [14,18]. Compared to salt-water droplets which may evaporate in a depinning mode depending on the salt concentration and wettability of the substrate [21–23], biological fluid drops generally evaporate in a constant contact line regime, because of the presence of proteins. During the drying of colloidal drops, surface tension gradients arise, due to local temperature and/or concentration variations

induced by evaporation of water. These lead to convection and mixing and, in turn, redistribution of the components. For biological fluid drops, water evaporation leads to increasing solute concentration near the contact line and, hence, to the development of a concentration gradient between the contact line and the centre. This is a result of the higher evaporation rate at the contact line, which induces supersaturation of components near the periphery of the drying drop. As a result of supersaturation near the periphery, preferential precipitation of the components occurs at the contact line, leading to the formation of an outer ring, mainly composed of high molecular weight components such as proteins. Protein adsorption takes via a series of adsorption-displacement steps in which proteins of lower molecular weight adsorb to the substrate first, and are subsequently displaced by proteins of higher molecular weights. This phenomenon is called the "Vroman effect" [24]. Low molecular weight components, on the other hand, typically precipitate in the centre of the drying drops [24]. The increase in the salt concentration during evaporation gives rise to surface tension gradients, affecting the phenomena occurring. The wetting behaviour of salt crystals significantly affects the growth and morphology of the formed patterns [25]. The evaporative dynamics, as well as the morphology of the final desiccation patterns, depend on the composition of the biological fluid [26]. Additionally, studies have attributed the ring formation to temperature variations that lead to surface tension gradients on the free surface of the drying droplets, due to evaporative cooling. Hu and Larson showed that reverse temperature gradients may occur when the initial contact angle changes from $40°$ to $10°$ [27]. Other studies have shown that the evaporation temperature affects the assembly mechanism of the particles in the drying droplets, with higher evaporation rates leading to irregular and more disordered assemblies [28,29].

Understanding of the mechanisms occurring during drying of bio-fluid drops, as well as of the final desiccation patterns, is crucial for their application to medical diagnostics and forensic analysis [30–33]. The stages of blood drop drying were reported by Sobac and Brutin [15]. Variations in the composition of biological fluids, caused by disease, have been found to alter the final dried deposits [34,35]. Brutin et al. [31] showed that desiccation patterns differ for blood drops acquired from healthy, anaemic and hyperlipidaemic people. Yakhno et al. [32] showed that the final dried patterns of serum drops acquired from healthy individuals reveal different morphologies compared to breast and lung cancer patients, and paraproteinemia patients. Additionally, Yakhno et al. [36] reported that desiccation patterns of serum drops acquired from healthy individuals differ than those acquired from viral hepatitis B and burn disease patients [36].

The acquisition of physiological fluids is challenging due to safety and storage issues. For this reason, many researchers have focused on the study of similar but simpler model systems such as inorganic [19] or organic [11,37] colloids and colloidal solutions with salt admixtures. Rapis [35] has investigated the self-organisation phenomenon occurring in a desiccating aqueous protein solution. A study has examined the evaporation of BSA (Bovine Serum Albumin) and lysozyme solution drops of different concentrations, reporting that the type and initial concentration of proteins in the drops determine the evaporative behaviour and the final pattern morphology [38]. Additionally, another work has examined the evaporation of aqueous bio-colloidal solutions containing liquid crystals and different types of proteins: BSA, lysozyme and myoglobin. Image processing techniques were employed in combination with cross-polarised and bright-field microscopy for the examination of the pattern morphology and the quantification of cracks [39].

The evaporative dynamics and the final desiccation patterns of bio-fluid drops depend on various factors that include the type and concentration of macromolecules and electrolytes [40,41] in the solution, the substrate properties such as wettability [11,31,42], and the ambient conditions at which drying takes place [43,44]. Studies have examined the effect of these factors on the desiccation of bio-fluid drops. Iqbal et al. [45] probed the effect of dilution and substrate wettability on the evaporative behaviour and desiccation patterns of blood drops. Additionally, Pal et al. [46] investigated the drying dynamics and morphological features of blood and aqueous BSA drops at various initial concentrations

and substrate temperatures. Very recently, Carreon et al. [47] examined the effect of substrate temperature on the drying patterns of protein mixtures and salt-protein mixtures, to investigate the most suitable drying temperature for efficient diagnosis. These studies provided a basis for the understanding of the effect of substrate temperature on the drying of protein solutions, protein–protein and protein-salt mixtures. However, because of the complexity of biological fluids such as blood and serum, further research is required for the understanding of the mechanisms implicated in the drying of such systems.

To the best of our knowledge, the impact of substrate temperature on the drying of serum drops has not been investigated yet. Blood serum is a very complex mixture consisting of various types of proteins, electrolytes, antibodies, antigens and hormones. Additionally, in previous studies examining the effect of temperature on the drying of protein-salt mixtures, the pH of the solutions was not controlled. The pH of the solution plays a major role in protein folding and aggregation, by affecting the electrostatic repulsive and attractive van der Waals forces between macromolecules. As a result, the pH of the solution significantly affects the final desiccation patterns which are governed by the interaction between the components of the solution. In this work, we use Foetal Bovine Serum (FBS), due to its similarity to human serum, to study the effect of substrate temperature on the dryout and deposits from drops. The temperature range of 20 °C to 40 °C was investigated, as it is the most relevant in diagnostic applications. Compared to other studies that focused on image analysis of protein-salt mixtures using image statistics, we provide topographical examinations of the dried deposits and attempt to correlate the drying dynamics to deposit topography and protein conformation.

## 2. Materials and Methods

Foetal bovine serum (FBS South American—A3160802, Thermofisher Scientific, Renfrewshire, UK) was received frozen in dry ice and used after defrosting and gentle mixing at room temperature. No vortex mixers or sonicators were used, to avoid protein denaturation. FBS is a complex solution consisting of multiple proteins, ions and hormones. The protein content of FBS is given in Table 1. The ionic strength and pH of the solution were ~0.14 M and 7.4, respectively.

**Table 1.** The type and concentration of proteins in the FBS used in the experiments.

| Type of Protein | Typical Concentration Range | Concentration in Our Sample |
|---|---|---|
| BSA | 17–35 mg/mL | 23 mg/mL |
| α-globulin | 7–20 mg/mL | 16 mg/mL |
| β-globulin | 3–9 mg/mL | 3.6 mg/mL |
| γ-globulin | 10–200 μg/mL | 23.73 μg/mL |
| Haemoglobin | 0.01–0.30 mg/mL | 0.1401 mg/mL |

Glass microscope slides (MS/1 Scientific Glass Laboratories Ltd., Stoke-on-Trent, UK) were placed in an ultrasonic bath containing deionized water for 15 min, rinsed with ethanol (Ethanol 99%+, Absolute, Fisher Scientific, Loughborough, UK) and dried using an air gun. Sessile drops of FBS (1.2 ± 0.2 μL) were gently placed on the glass slides and left to dry at different substrate temperatures. The initial contact angle between the liquid drops and the substrates was 35 ± 5°. The substrate temperature was controlled (PID control) via the use of a heater mat (SRFRA-4/10-230V, Omega Engineering Ltd., Manchester, UK), placed underneath the glass surfaces, and a thermocouple mounted on the slide surface (Figure 1). Both the ambient temperature and the relative humidity (RH) levels were monitored during the experimental procedure via the use of a temperature-humidity meter (HH311, Omega Engineering Ltd., Manchester, UK). RH levels were 45% ± 5% during the experiments, and the ambient temperature was 20 ± 1 °C.

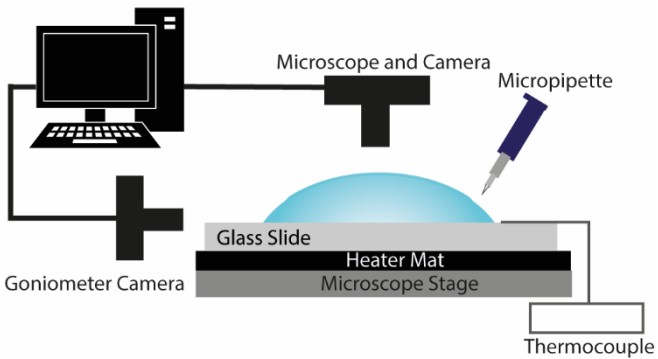

**Figure 1.** Schematic diagram of the experimental setup.

A goniometer (DSA-30S Drop Shape Analyser, KRÜSS, Hamburg, Gemany) enabled side-view imaging of the drying drops and was used to capture the evolution of the droplet volume, diameter and contact angle over time. Additionally, for droplets drying at 20 °C, the temporal evolution of mass was monitored via the use of a microbalance (Gemini, GR-202, A&D, Japan). The acquisition of mass measurements for droplets drying at higher substrate temperatures was not possible due to limitations of the experimental apparatus.

Top view images of the drops were captured under $2\times$ magnification during drying via the use of an optical microscope (Euromex, Arnhem, The Netherlands) connected to a CMOS camera (MAKO G-507 Allied Vision Technologies, Farnham, UK). Top view images were also acquired after the end of drying under different magnifications ($2\times, 5\times, 10\times$) for the investigation of the final desiccation patterns. The dried deposits were also examined 24 h after completion of the experiments.

Three-dimensional topographical studies were conducted on the desiccation deposits, under $20\times$ magnification, via the use of a confocal laser scanning microscope (VK-X1000, Keyence, Eindhoven, The Netherlands) in combination with Keyence Multifile Analyzer software. Because viewing of the entire desiccation pattern was challenging under such high magnification, different regions in the dried deposits were examined separately, and then stitched together in Keyence Multifile Analyzer, to provide the topography of the final desiccation patterns. The experimental procedure was similar to that described above. Topographical studies allowed the measurement of the average thickness (height) and radius of the dried deposits, as well as measurement of the average thickness and length of each zone within the deposits. This enabled the calculation of the final deposit volume and the volume of each zone in the dried deposits, based on thickness and area.

Each set of experiments was repeated at least three times for each substrate temperature. The morphology and topology of the final desiccation deposits showed good reproducibility at each substrate temperature in all of these methods.

## 3. Results

### 3.1. Evaporation Stages

When a small drop, with a size below the capillary length, of serum is placed on a glass surface ($h \ll R$), it acquires a spherical-cap shape. In this case, gravitational forces can be neglected, because of the small Bond number ($Bo = \frac{\rho g h^2}{\gamma} \sim 0.3$ at 20 °C). Following drop's spreading on the substrate, evaporation takes place, leading to desiccation, which gives rise to distinctive patterns. Irrespective of the temperature, the desiccation process can be divided into four stages. These are: pre-gelation, gelation, crystallisation and crack formation.

Figure 2 shows the temporal evolution of the normalised volume, contact angle and height, during the desiccation process. The evolution of volume is shown until the end of cracking, which denotes the end of drying. However, during this stage the volume is low and the resolution of the apparatus is not high enough to track any changes. Volume, contact angle and height were normalised based on the initial values at the onset of drying.

Additionally, Figure 3 shows the temporal evolution of the normalised mass during drying, for FBS drops drying on glass slides of 20 °C. However, because of the nature of the heating apparatus, mass measurements could not be acquired for higher substrate temperatures.

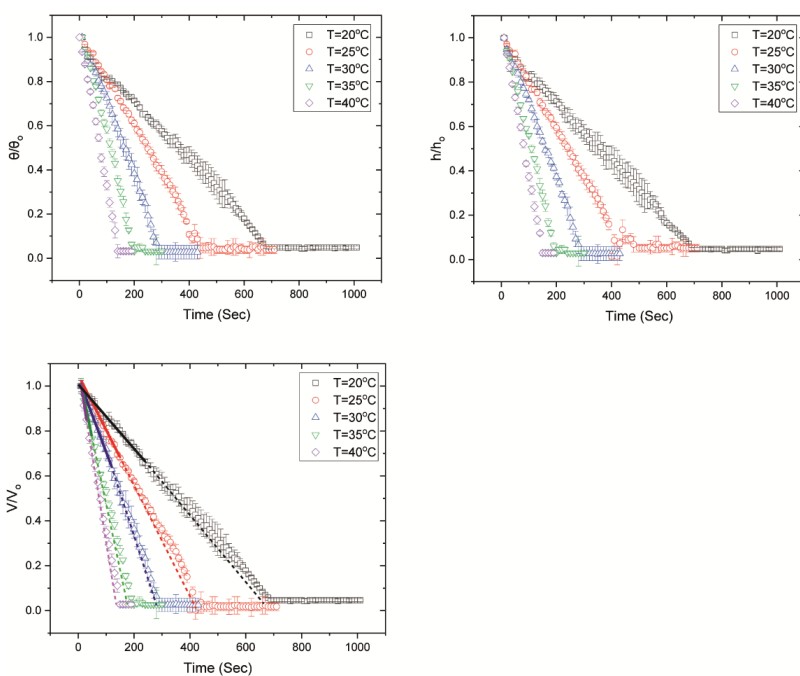

**Figure 2.** Temporal evolution of the normalised contact angle ($\theta/\theta_o$), height ($h/h_o$) and volume ($V/V_o$) for FBS drops evaporating at different substrate temperatures. The volume decreases linearly indicating an almost constant evaporation rate until the onset of gelation (continuous lines) [48,49], when the evaporation rate decreases slightly (dashed lines). The volume remains constant throughout crystallisation and cracking. We show the volume during crystallisation and cracking, although the resolution of the experimental apparatus is not high enough to track volume changes during these stages. It should be noted that, because the volume is estimated from the side profile of the drying drops, there is an overestimation of the volume after the formation of the peripheral ring. The average times for the onset of gelation are ~250 s, ~163 s, ~129 s, ~66 s and ~38 s for drops evaporating on glass slides with temperatures of 20 °C, 25 °C, 30 °C, 35 °C and 40 °C, respectively.

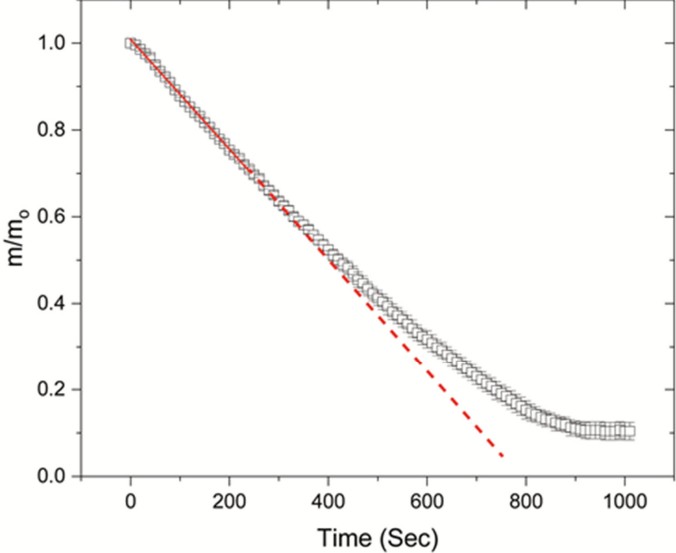

**Figure 3.** Temporal evolution of normalised mass ($m/m_o$) for FBS drops drying on glass slides of 20 °C.

### 3.1.1. Pre-Gelation

In the first stage (up to ~25 ± 5% of the droplet lifetime), the droplet is liquid and desiccation is dominated by the evaporation of water from the droplet edge, where the evaporation rate is higher [1,16]. Due to the higher evaporation rate at the contact line, fluid is transferred towards the periphery via a radially outward capillary-driven flow, leading to aggregation of the macromolecular proteins near the periphery, and hence, pinning of the drops to the substrate. The pinning of the contact line to the glass slide leads to a constant contact radius mode of evaporation with a decreasing contact angle as drying proceeds. During this stage, the droplet volume decreases in an almost linear manner (Figure 2), indicating that the evaporation rate is constant, until the onset of gelation. This is similar to what has been observed for the evaporation of a pinned pure solvent drop [50,51].

### 3.1.2. Gelation

The accumulation of proteins at the peripheral region is followed by the gelation process (Figure 4), during which a sol-gel transition occurs. Gelation dominates the second stage of the evaporation process, commencing at the periphery of the droplet and propagating towards the centre that is still in a liquid state. There was a variation regarding the times that gelation lasted, from 25–30% to 60–80% of the droplet's lifetime. During this stage, the width of the peripheral ring increases with time. The formed gel is a porous film that traps water molecules. Water evaporation continues after the formation of the gelation film, but at a slower rate (Figure 2) since water molecules have to diffuse through this film to escape through the interface to vapour [52].

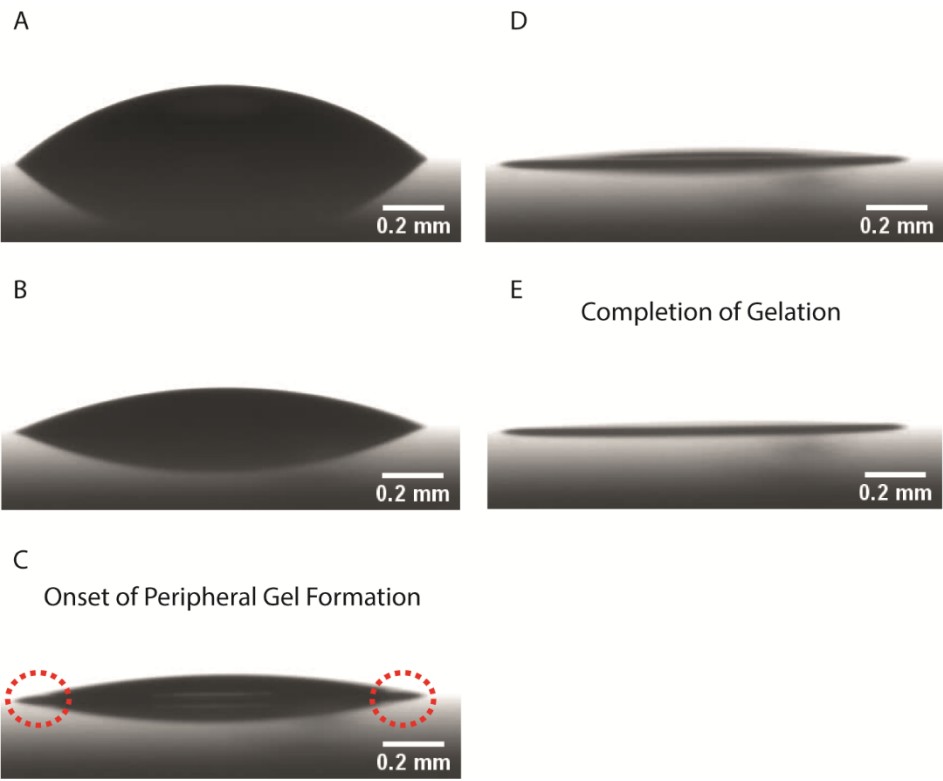

**Figure 4.** Side-view images acquired from ADVANCE software showing the stages of evaporation and gelation of a drop evaporating at 25 °C. (**A**) Immediately after droplet deposition on the glass slide, (**B**) 37% of the time for complete gelation (215 s), (**C**) 57% of the time for complete gelation when the peripheral protein ring forms (327 s), (**D**) 70% of the total gelation time (394 s) and (**E**) complete gelation of the drop (575 s).

### 3.1.3. Crystallisation and Crack Formation

Subsequent to the completion of gelation, crystallisation takes place in the central region of the droplet. In all the examined cases, the propagation of the crystals is very rapid initially and subsequently slows down. Along with the crystallisation process, cracks form on the peripheral protein ring, propagating towards the crystalline central region of the droplet. Crack formation commences on the peripheral gel almost simultaneously with the nucleation of crystalline features in the central region (exactly upon crystal nucleation or within a few seconds). It is worth noting that crack formation continues after the completion of crystallisation. No significant volume changes are observed during crystallisation and cracking (Figure 2).

It should be noted that, once the peripheral ring is formed, the estimated volume appears to be constant, although evaporation takes place from the centre of the drops. Because of the ring formation during gelation, which is thicker than the central region of the deposits, there is an overestimation of the volume (determined from the side profile) presented in Figure 2, during the final drying stages. Additionally, Figure 3 illustrates that the temporal evolution of mass is not linear after the onset of gelation, because of the formation of the gel. In all cases, the final volume of the deposits is not zero, but it drops to the order of tens of nanolitres.

The stages described here are similar to those observed by Chen et al. [41] for plasma drops derived from healthy adults, doped with different initial NaCl concentrations. However, here we describe the first stage as pre-gelation instead of solution. It is noteworthy that in their work on human plasma drops, Chen et al. [41] observed the onset and completion of crack formation prior to the onset of crystallisation. In our work for FBS drops, crystallisation and crack formation occur almost simultaneously in most of the examined drops, with crack formation lasting longer, until the complete desiccation of the drops.

The evaporation rate increases with increasing substrate temperature and can be described by a second order polynomial trend, as shown in Figure 5. The evaporation rate should be proportional to the driving force concentration difference ($\Delta c$) which may be described in terms of mole fraction, molarity or partial pressure [53]. In our case, the evaporation is driven by vapour pressure through diffusion, hence, the driving force is the pressure difference ($\Delta p$). For any substrate temperature, during pre-gelation, the evaporation rate should be proportional to the pressure difference $\Delta p$, which was calculated as $P_{sv} - P_v$, where $P_{sv}$ is the saturated vapour pressure of pure water at each examined substrate temperature and $P_v$ is the partial pressure of water due to relative humidity at the same temperature. Relative humidity ($RH$) is given by:

$$RH = \frac{P_v}{P_{sv}}$$

The values of saturated vapour pressure for pure water ($P_{sv}$) with temperature are given in Table 2. Based on the RH levels under which the experiments were performed, the partial water pressure ($P_v$) may be estimated. It should be noted that this is an approximation, as we do not consider the changes of water vapour pressure due to the presence of proteins. The vapour pressure change with temperature is given in Figure 5. Figure 6 shows the evaporation rate with vapour pressure, along with images of the final desiccation patterns for each examined substrate temperature. The evaporation rate presented in Figure 6 was calculated for the pre-gelation stage, when the evaporative behaviour is similar to that of a pure liquid and concentration effects are not significant within the drying drops. The change in the evaporation rate is linearly proportional to the pressure difference, suggesting that during pre-gelation the evaporation is driven by the vapour pressure of pure water.

**Table 2.** Saturated vapour pressure of water with temperature.

| Temperature (°C) | Saturated Vapour Pressure of Water ($P_{sv}$) (kPa) |
|---|---|
| 20 | 2.33 |
| 25 | 3.17 |
| 30 | 4.24 |
| 35 | 5.63 |
| 40 | 7.38 |

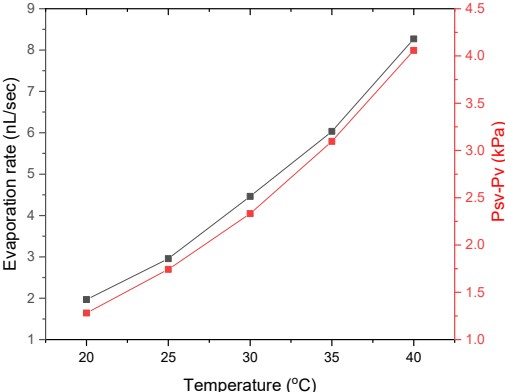

**Figure 5.** Evaporation rate during the pre-gelation stage and vapour pressure changes with substrate temperature.

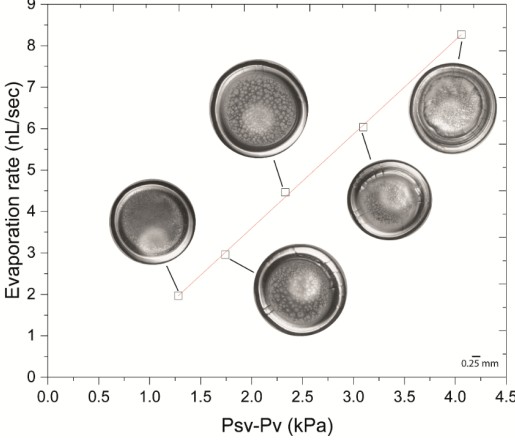

**Figure 6.** Evaporation rate with vapour pressure. The evaporation rate during pre-gelation is linearly proportional to the vapour pressure change, suggesting that drying is driven by vapour pressure. Images show the final dried deposits after drop evaporation at each examined substrate temperature.

*3.2. The Effect of Temperature on the Morphology and Topography of the Final Desiccation Deposits*

Yakhno et al. [34] showed that four distinct zones exist in the desiccation patterns of aqueous 7% BSA-0.9% NaCl drops. Moving from the deposit periphery towards the centre, these zones are: (1) a (glassy) peripheral ring of homogeneous protein, (2) a zone of protein structures, (3) a protein gel, and (4) a crystalline zone (Figure 7). Proteins exist at different conditions in each zone, forming materials of different properties; a glassy (high volume fraction) protein ring on the periphery, and a protein gel (lower volume fraction) in the interior of the ring. In the central area of the desiccated drops, the volume fraction of proteins is lower, but the ionic strength increases [34]. We observe the same zones in the dried deposits of FBS drops, where the gel zone is larger at higher temperatures.

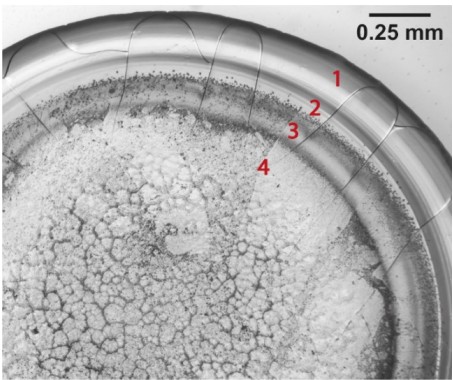

**Figure 7.** Four distinct zones observed in the desiccated patterns. (1) The peripheral protein ring formed on the drop periphery where cracking takes place, (2) clusters of protein structures, (3) the protein gel and (4) the crystalline area consisting of finer structures. The image was acquired after desiccation of a FBS drop at 40 °C under 5× magnification, 24 h after the completion of the experiment.

The topography of the dried deposits was investigated using confocal laser scanning microscopy (Keyence VK-X1000). Figure 8 shows 3D representations of the final deposits for each examined substrate temperature.

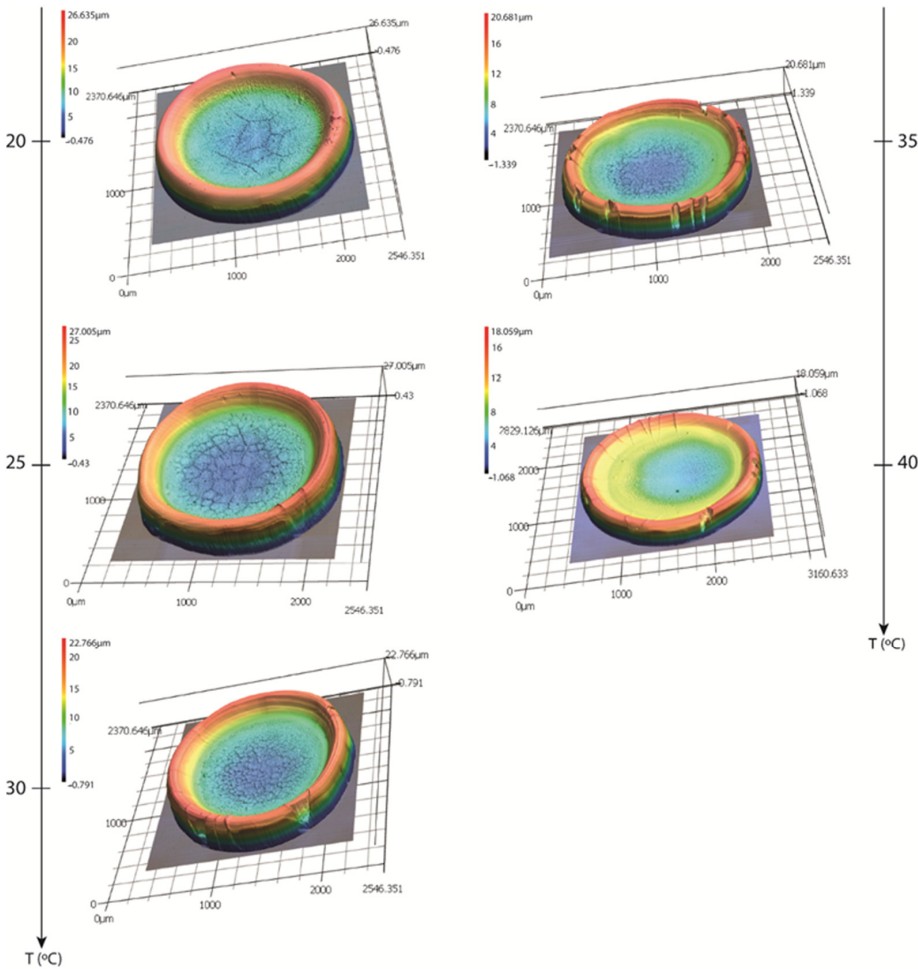

**Figure 8.** Three-dimensional representation of the final desiccation deposits occurring after the evaporation of the FBS drops at different temperatures.

## 4. Discussion

The peripheral protein ring is the tallest feature in the desiccation patterns, for all of the examined substrate temperatures. Topographical examination of the final patterns has shown that the deposit height decreases from the periphery towards the centre. Figure 9 shows the zones of the ring (red), protein structures with the protein gel (yellow and green) and crystalline structures (blue).

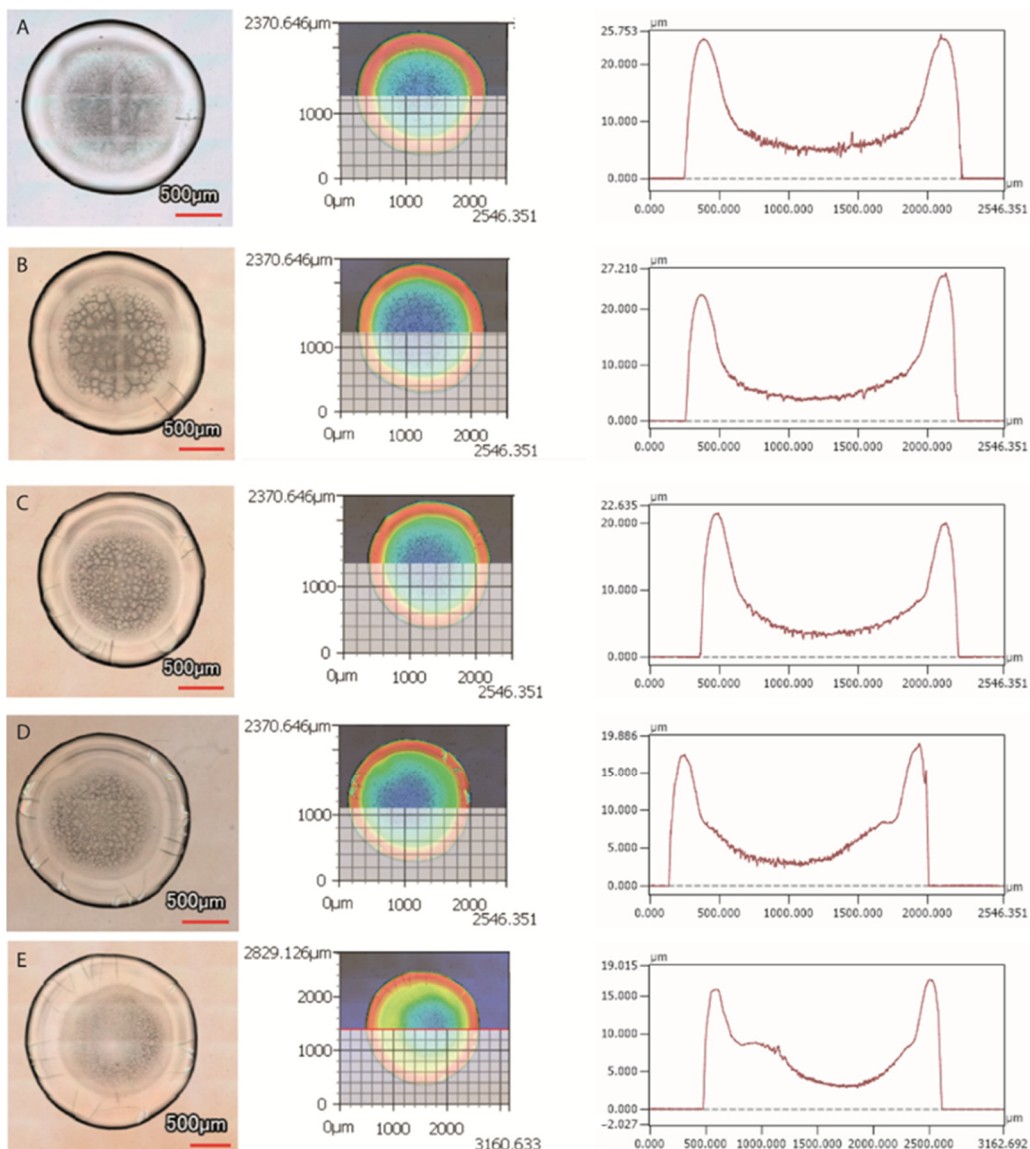

**Figure 9.** Top view-images of the dried deposits (**left**), illustration of the horizontal line along which the height profile is drawn (**middle**), and height profile of the deposits occurring after drop evaporation (**right**) at (**A**) 20 °C, (**B**) 25 °C, (**C**) 30 °C, (**D**) 35 °C, and (**E**) 40 °C.

Additionally, for drops drying at substrate temperatures of 20 °C to 35 °C, the volume of the final deposit decreases (from 44.8 nL to 24.4 nL) with increasing substrate temperature (Figure 10). This may indicate that the deposits formed at higher temperatures have a lower water content, compared to those formed at lower temperatures. Nonetheless, we should note that this is not the case when the substrate temperature increases from 35 °C to 40 °C, as an increase is observed in the average volume of the final deposits (~35.6 nL). It is possible that this indicates conformational changes of proteins. BSA, which is the most abundant protein in FBS, typically undergoes irreversible conformational changes at temperatures exceeding 58 °C. Nevertheless, studies have found that certain types of

proteins, or protein mixtures, can denature at lower temperatures. It has been found that bovine haemoglobin undergoes reversible structural changes for temperatures above 35 °C and pH ~6–8. Temperatures higher than 35 °C lead to a decrease in the α-helix content of haemoglobin, caused by breakage of the stabilising hydrogen bonds [54]. As a result, the flexibility of the protein molecule increases, leading to exposure of the side chain groups to the solvent. Reversible denaturation at such temperatures was also found to occur for haemoglobins of other species [55], including human haemoglobin [56]. Digel et al. [55] showed that reversible conformational changes occur for temperatures near the species' body temperature. It is noteworthy that BSA is also subject to reversible conformational changes upon temperature increasing from 25 °C to 40 °C which also manifest as a decrease in the α-helix content of the molecule [47]. Additionally, the presence of ions in FBS has an impact on protein structure, affecting the denaturation temperature. It is noteworthy that drying also affects the structure of proteins, as water evaporation enhances dehydration and increases the ionic strength in the drying drops, leading to irreversible denaturation and aggregation. Therefore, the increase in the average deposit volume from 35 °C to 40 °C may be a result of conformational changes induced by temperature and increasing ionic strength. Temperature also has an effect on the self-assembly mechanism of molecules. Lower evaporation rates, exhibited at lower substrate temperatures, allow more time for the ordering and assembly of molecules during drying, whereas for higher substrate temperatures, when the evaporation rate is higher, the assembly of molecules may be irregular and more random [28,29]. The assembly mechanism of molecules can, therefore, affect the topography of the final deposits. In the case of our experiments, the abrupt change in the final deposit volume in a small temperature range of 5° (from 35 °C to 40 °C) would suggest that protein denaturation is the most plausible mechanism for the topographical changes observed.

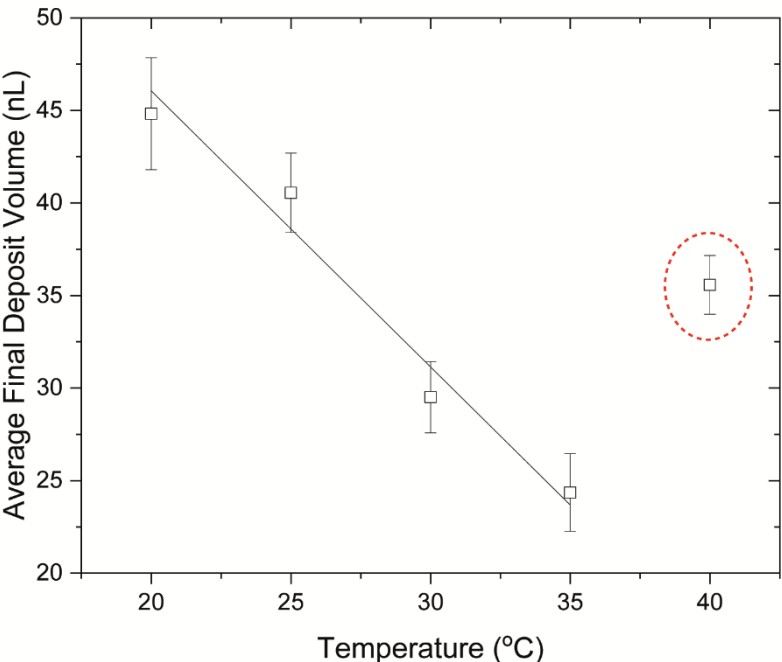

**Figure 10.** Average volumes of the final desiccation deposits based on the substrate temperature during evaporation. The dashed ellipse shows the anomaly observed at 40 °C, when the volume of the final deposit increases.

The volume of each zone in the desiccation patterns was calculated. It should be noted that the protein structures were considered as part of the gel for the volume calculation. For the crystalline zone, the total volume of the zone was calculated, including the crystalline structures and the underlying protein gel. Figure 11 shows the volume of each zone in

the dried deposits with temperature. The volumes of the peripheral ring and the central crystalline zone decrease from 20 °C to 35 °C, whereas an increase of the volume is observed for both zones when the substrate temperature increases from 35 °C to 40 °C. For the gel, the lowest volume is observed at 20 °C and an increase is observed at 25 °C. Between 25 °C and 35 °C no significant volume changes of the gel are observed. However, at 40 °C, the gel volume is approximately two times higher than that between 25 °C and 35 °C.

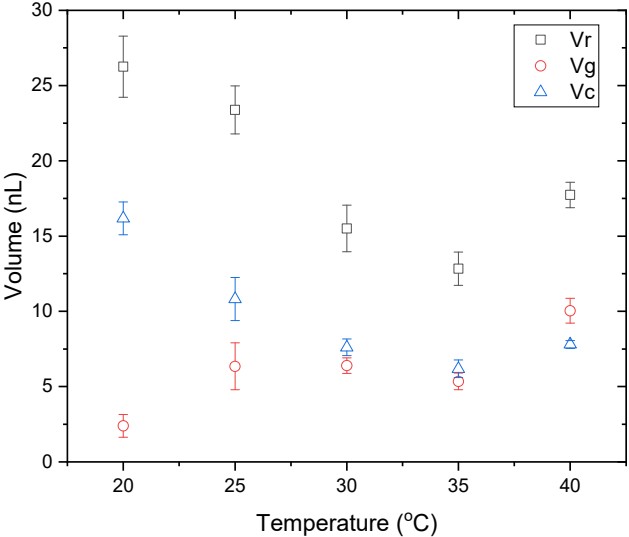

**Figure 11.** Final volumes of the ring ($V_r$), gel ($V_g$) and crystalline zones ($V_c$) in the desiccation deposits with temperature.

*Glassy Peripheral Protein Ring and Crack Formation*

The glassy peripheral protein ring formed at the edge of the deposit, with an almost constant width for all of the examined temperatures (0.15–0.25 mm). Previous works have reported that the width of the protein ring increases with increasing initial protein concentration [57]. In our case, the FBS was used as is, therefore the initial protein concentration was the same in all the examined cases. However, it should be noted that large variations in the evaporation temperature can affect the final width of the peripheral ring [58]. Recently, Pal et al. [46] showed that the width of the peripheral ring changes as the substrate temperature increases from 25 °C to 55 °C for dried deposits of BSA drops. However, in our case, no significant changes of the ring width were observed.

The peripheral ring is an elastic and deformable gel which entraps water molecules [14]. At the final stages of desiccation, when evaporation of the fluid from the gel phase takes place, the gelled film starts to shrink. The adhesion of the film to the substrate prevents the shrinkage, causing internal stress accumulation. The accumulation of stress, in turn, causes an increase in the stored strain energy [52]. As a result of the increasing strain energy, when the stress exceeds a critical value, cracks are formed on the peripheral protein ring [59]. Crack formation indicates the stress release caused by the competition between drop adhesion to the substrate and the internal stresses that develop during desiccation [16]. Two types of tensile stress may act, leading to the formation of either radial or orthoradial cracks. In both cases, the stress acts normal to the direction of the formed cracks (Figure 12). Crack formation is initiated with the appearance of radial cracks within the drying drops, indicating that the stress predominantly acts in the orthoradial direction. The formation of radial cracks is followed by the development of tangential cracks, branching out from the already formed radial cracks.

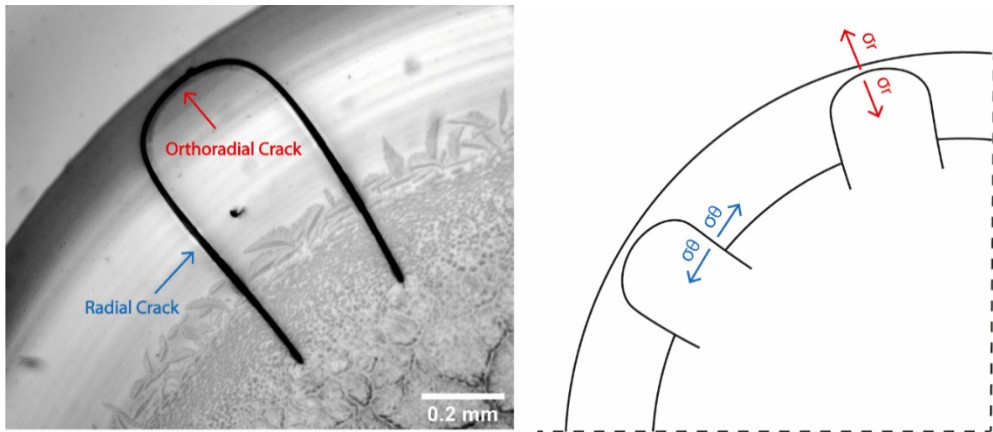

**Figure 12. Left**: Radial and orthoradial cracks on the desiccation deposit of a droplet dried on a glass slide with temperature T = 25 °C (10× magnification). **Right**: Schematic showing the stresses in the orthoradial direction (blue) leading to the formation of radial cracks near the periphery of the drying drop and the development of radial stresses (red) leading to the formation of orthoradial cracks.

In what follows, we employ the Griffith theory of fracture and attempt to explain our observations on the cracking of the final desiccation deposits. According to Griffith theory, the critical cracking stress for the formation and propagation of a narrow straight crack is given by:

$$\sigma_c = \sqrt{\frac{G_c E}{\pi \alpha}} \tag{1}$$

where $E$ is the elastic modulus, $G_c$ is the critical strain energy release rate and $\alpha$ is the length of the crack [52,60,61]. At this specific stress, crack activation occurs with subsequent propagation. The stress is caused by the strain energy that is stored in the neighbouring area of a microscopic defect [60]. The critical strain energy release rate $G_c$ represents the rate below which the crack does not release enough strain to propagate. When the energy release rate $G$ is higher than $G_c$, on the other hand, crack propagation takes place. However, when $G = G_c$, an equilibrium is reached and the crack neither propagates nor shrinks [61]. Additionally, a critical crack length $\alpha_c$ exists, activated by a far-field stress ($\sigma_0$), which determines whether existing cracks will remain inactive. Substitution of $\sigma_0$ in Equation (1) and rearranging gives the critical crack length as [52,61]

$$\alpha_c = \frac{G_c E}{\pi \sigma_0^2} \tag{2}$$

Nonetheless, as soon as the critical length value is reached, crack propagation takes place [52].

Since the bonds near the tip are involved in crack propagation, any stress concentration causes the gelled film to weaken globally. The distribution of stress around the tip of a crack is given by:

$$\sigma_{tip} \sim 2\sigma_0 \sqrt{\frac{\alpha}{R}}, \tag{3}$$

indicating that as evaporation proceeds and more water evaporates, the stress becomes more concentrated and crack propagation takes place [61].

These results were confirmed by Bonn et al., who showed and explained a delayed fracture in inhomogeneous soft polymer gels [62].

In our experiments, an increased number of cracks was observed with increasing substrate temperature (Figure 13). Focusing on the radial cracks, for the drops evaporating at 20 °C, one or no radial cracks were formed. The highest number of cracks occurred at the maximum examined temperature, i.e., 40 °C (with an average of ~16 cracks), when the evaporation was the fastest.

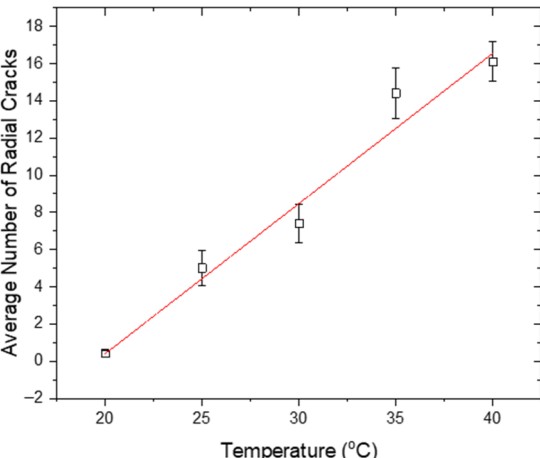

**Figure 13.** Average number of radial cracks occurring on the drop periphery after desiccation, plotted as a function of temperature.

The height investigation demonstrated that the rings of lower heights, which formed at higher drying substrate temperatures, manifested a larger number of cracks for the release of the accumulated stress. This agrees with the findings of Lama et al. [59], claiming that for lower height deposits, the number of cracks is higher. The final average ring height decreases with increasing temperature from 20 °C to 35 °C (from 22.2 μm to 12.8 μm). However, when the substrate temperature increases from 35 °C to 40 °C, the final average ring height increases to ~14.8 μm (Figure 14). This may be a result of protein unfolding as already discussed.

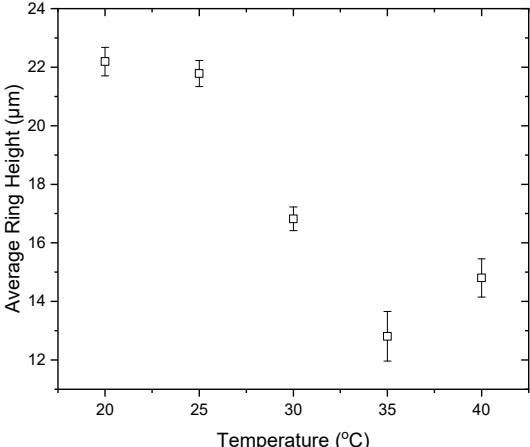

**Figure 14.** Average ring height in the desiccation patterns acquired after droplet drying at different substrate temperatures.

Furthermore, as mentioned above, the final volumes of the desiccation deposits formed at higher substrate temperatures (from 20 °C to 35 °C) are lower compared to those formed at lower temperatures, a phenomenon that reverses in the range from 35 °C to 40 °C. In the former case, since water is the only component that evaporates during desiccation, a lower final volume may indicate lower water content within the deposit. Thus, the gelation film formed at higher temperatures could be more brittle [52] and could explain the larger number of cracks appearing on dried deposits formed at higher temperatures. Nonetheless, this is not the case when the substrate temperature is 40 °C. In that case, an increase is observed in the volume of the final deposit, probably due to protein unfolding. Protein unfolding may lead to increasing strain and, hence, increasing stress in the drying drops, leading to a higher number of cracks for stress release.

Additionally, the evaporation rate increases at higher temperatures because of higher vapour pressure of water with temperature, enabling faster water evaporation which results in a faster reduction in the droplet height, as shown in Figure 2. Because of enhanced evaporation at higher temperatures, as the entrapped water molecules escape the porous medium, higher internal stresses develop within the drying drop, causing a higher number of cracks to appear on the periphery [41]. This may be due to a higher rate of water diffusion and evaporation from the gel. At higher temperatures, the diffusion coefficient increases, enhancing the diffusion and evaporation of water molecules, leading to faster surface evaporation. This allows more water molecules to escape to vapour before the gel has reached its final form, increasing the internal stress within the gel. For the drops evaporating at lower temperatures, on the other hand, a longer time is required for the water trapped within the gel to evaporate, which slows down the evaporation, minimising the stress and hence, the number of cracks [41]. Additionally, the higher number of cracks in deposits formed at higher substrate temperatures may be attributed to a lower water content in the deposits. For drops dried on substrates of higher temperature, the final volume of the deposits decreases, indicating a lower final water concentration in the gel. This indicates a more brittle gel, favouring crack formation.

## 5. Conclusions

The evaporation of biological fluid drops gives rise to distinct desiccation patterns which may be used as a tool for disease diagnosis. The substrate temperature during drying affects the evaporation process and the final desiccation deposits. In this work, we studied the evaporative dynamics of FBS drops drying on glass slides of temperatures between 20 °C and 40 °C, as well as the topography and morphology of the final desiccation deposits. This temperature range was chosen as it is the most relevant for diagnostic applications. The evaporation process is divided into four distinct stages for all the examined substrate temperatures; pre-gelation, gelation, crystallisation and crack formation. These findings agree with the findings of Chen et al. [41] for desiccation patterns of human plasma drops doped with different NaCl concentrations. Nonetheless, in our case, crystallisation and crack formation occur almost simultaneously and crack formation lasts longer, until complete desiccation of the drops. The final deposits consist of four regions with different characteristics: a peripheral ring of homogeneous protein, a zone of protein structures, a protein gel and a central crystalline zone. The final average volume of the deposits and the protein ring decrease with increasing substrate temperature from 20 °C to 35 °C, indicating a lower water content for deposits formed at higher temperatures. However, the volume increases when the substrate temperature increases from 35 °C to 40 °C, which may indicate protein denaturation at 40 °C. We focused on the peripheral ring showing that the average ring height decreases with increasing temperature from 20 °C to 35 °C, whereas the number of cracks increases. This agrees with the findings of Lama et al. [59], reporting that for deposits of lower height, the number of cracks per unit length is larger. Nevertheless, in our case, the average ring height increases with increasing substrate temperature from 35 °C to 40 °C, and a larger number of cracks forms on the periphery. The findings of this study suggest that for substrate temperatures between 20 °C and 35 °C, cracking is governed by the evaporation rate and the final water content in the deposits; however, at 40 °C, protein denaturation should also be taken into account.

The findings of this work could contribute to the understanding of the impact of environmental conditions on the desiccation of biological fluid drops as a tool for disease diagnosis and forensic applications, as patterns may be affected by the temperature at which desiccation takes place. Investigation of the FBS droplet evaporation at a larger range of temperatures may be required for a better understanding of the factors affecting the final desiccation patterns; however, this was beyond the scope of this study.

**Author Contributions:** The manuscript was written through contributions of all authors. Conceptualization, M.E. and J.C.; methodology, M.E.; software, M.E.; validation, M.E. and J.C.; formal analysis, M.E.; investigation, M.E.; resources, J.C., K.S. and D.B.; data curation, M.E.; writing—original draft

preparation, M.E.; writing—review and editing, J.C., K.S. and D.B.; visualization, M.E.; supervision, J.C. and K.S.; project administration, J.C. and K.S.; funding acquisition, J.C. and K.S. All authors have read and agreed to the published version of the manuscript.

**Funding:** This research was funded by the Newton Fund, grant number 337066.

**Data Availability Statement:** The data presented in this study are available from the authors upon request.

**Acknowledgments:** The authors would like to thank Paul Kolpakov (Van der Waals-Zeeman Institute, Faculty of Science, University of Amsterdam) for the training on Confocal Laser Scanning Microscopy.

**Conflicts of Interest:** The authors declare no conflict of interest.

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
