# Peer review of "The Effect of Substrate Temperature on the Evaporative Behaviour and Desiccation Patterns of Foetal Bovine Serum Drops"

_colloids, doi:10.3390/colloids5040043_

Round 1

Reviewer 1 Report

The authors carried out an interested investigations of the evaporation of biological fluid drops, and the effects of temperature on the ‘coffee-ring’ formation. It is meaningful for disease diagnosis, nanoparticles self-assembly and nanosuperstructures fabrication. I like to recommend this paper to publish in Colloids Interfaces after the following considerations.

- it may figure out a density curve supplemented to Figure 2.

- I wonder how to determine the Pv ? more explanations should be given.

- the interested crystallisation processes are not described enough. However, these processes could involve in the most interested protain self-assemblies with long-range order. I strongly suggest they supplement further evidence of SEM results and give relevant descriptions.

- The formation of "coffee ring" effect is also ascribed to the thermal Marangoni flow and thermal disturbance, the authors need to explain the behaviors according to the theories, referring to the following works: Hu, H. et al., Langmuir 2005, 21, 3972−3980; Lin, Z. Q. et al., Angew. Chem. Int. Edit. 2012, 51, 1534−1546; Xie, Y. et al., Langmuir 2013, 29, 6232−6241.

Reviewer 2 Report

The authors have studied the drying process of small bio-fluid droplets, as well as the morphology of the resulting sediment of nonvolatile biocomponents at different substrate temperatures. This study shows how the competition between different mechanisms such as an increase in the evaporation rate with increasing temperature to a critical value and the simultaneous denaturation of proteins affects the structure of the final sediment. The results obtained are of great importance for the development of methods for medical diagnostics of diseases and forensic analysis.

The article is clearly written and well structured. The subject of the research corresponds to the journal's scope and requirements.

Remarks:

1) Figure 3 shows the temporal evolution of normalized mass for FBS drops drying on glass slides. However, the experimental part does not describe how the mass was measured? Did authors use a precise balance or calculated using the variation of volume over time of evaporation? If, by calculation, it is necessary to know the density of the solution at each moment of time, since the concentration of the components changes. It is necessary to provide explanations.

2) It is also not clear how the dependency shown in Figure 5 were obtained. If equations were used for the calculation, then it must be given in the text.

3) Please, describe the methodology for calculation/measurement of the average volumes of the final desiccation deposits (Figure 10) and the final volumes of the ring, gel and crystalline zones in the deposits (Figure 11).

That work can be published after revision.

Reviewer 3 Report

In this paper, the morphology and evaporation dynamics of the deposits produced by the evaporation of FBS droplets at different temperatures were studied. Such information is useful in the diagnosis of some pathologies and health problems and forensic analysis. Therefore, the authors investigated a diagnostic-relevant temperature range of 20–40 °C. The authors found that not only, the evaporation rate and water concentration impact the number of cracks per drop deposit, but also the protein denaturation when deposited at elevated temperatures. The presentation of the work is clear, the results are sound, and overall it was a pleasurable paper to study.

However, it is not completely convincing that no work so far has investigated the impact of substrate temperature on the drying of serum drops (line 104), especially when citing 44 and 45 references were the deposition of similar compositions (BSA+lysozyme+NaCl and BSA, respectively) were studied on a hot-stage. Authors should strengthen their point.

Figure 2. I had trouble noticing the continuous line at first. I suggest increasing thickness or highlighting both continuous and dashed lines.

Line 230. The incorrect reference to the figure is given in the text “Additionally, Figure 2 illustrates..”. I suppose it should be Figure 3.

Round 2

Reviewer 1 Report

I agree to publish the paper.